# Study on Chinese Farmland Ecosystem Service Value Transfer Based on Meta Analysis

**DOI:** 10.3390/ijerph20010440

**Published:** 2022-12-27

**Authors:** Liangzhen Nie, Bifan Cai, Yixin Luo, Yue Li, Neng Xie, Tong Zhang, Zhenlin Yang, Peixin Lin, Junshan Ma

**Affiliations:** 1College of Landscape Architecture, Zhejiang A & F University, Hangzhou 311300, China; 2College of Landscape Architecture, Nanjing Forestry University, Nanjing 210037, China

**Keywords:** ecosystem services, value transfer, farmland, meta-regression analysis

## Abstract

An analytic database was built based on meta-regression analysis (MRA) method, mainly including ecosystem service type, farmland division, cultivated land type, value assessment method, and farmland characteristics. The feasible weighted least square (FWLS) method was adopted to comprehensively investigate the seventy observations from empirical studies. The results indicate that: (1) except the negative impact of farmland area on farmland value, such factors as paddy field, good soil conservation function, mainly providing agricultural products, and using market value method for assessment all produce positive effect on the promotion of farmland value. (2) In meta-regression analysis, the average transfer error is 36.74%, and the median transfer error is 14.59%. (3) Under the A1B, A2, B1, and B2 scenarios of IPCC SRES, it is discovered from calculation that the value changes under different scenarios have some differences, in which, the total value rises significantly under A2 scenario and will reach to 15,220 billion yuan until the year of 2100; while the total value loss is the greatest under B1 scenario and will fall to 6320 billion yuan until the year of 2100. Finally, this paper gives some suggestions for scholars to deeply study the service value of farmland ecosystem as well as for the government to formulate differentiation policies.

## 1. Introduction

Nowadays, the increasingly severe natural resource depletion, ecology destroying, and environmental pollution have attracted people’s attention to the environment [1]. China is the world’s largest developing country, and its long-term extensive way of development has resulted in many ecological environment problems, which show sharp conflict to people’s growing demands for comfortable environment. Around the world, the farmland ecosystem is always one of the most important ecosystems [2]. Up to the 25 August 2021, according to the public data of Ministry of Natural Resources of the People’s Republic of China, Chinese cultivated land area is 1.43 × 10^6^ square kilometer, covering 14.90% of national territorial area, mainly distributed in Northeast China, Huanghai-Huaihai-Haihe region and Yangtze Plain, Middle, and Lower. Farmland ecosystem not only provides crops [3,4,5] and recreational tourism [3,6] for people, but also plays an important role in gas regulation [3,7,8], soil conservation [3,4,6,9], water conservation [3,10,11], and soil nutrient circulation [3,4], with huge environmental and economic benefits [12]. Recently, the main methods to evaluate farmland ecosystem service value include shadow project approach, market valuation approach, surrogate market approach, and carbon tax approach [11,13,14,15]. Before 2003, the ecosystem service value assessment in China mainly focused on forest [16,17,18], wetland [19], landscape [20], and river ecosystems [21,22], especially for forest and wetland ecosystems, but there were few contents about assessment of farmland ecosystem service value. In 2003, Zhao Rongqing [23] et al. initially defined and classified farmland ecosystem service functions. After that, China started to study the farmland ecosystem service value of different or specific region at different time periods [3,24,25].

At present, research on the application of Meta-analytic value transfer methods in the field of ecosystem service valuation is mainly two-fold: first, to develop a meta-analytic value transfer model with general applicability for ecosystem service value prediction within a broad ecosystem service valuation framework, and second, to develop a meta-analytic value transfer model for a specific policy context [26].The applied research of meta-analysis value transfer method in farmland ecosystem service value assessment field mainly focuses on the former aspect. In the research of ecosystem service value transfer, meta-analysis method can take effect in three points: (1) Integrate and evaluate the ecosystem service value assessment study; (2) build meta-regression model for evaluating the influential factors of ecosystem service value changes; (3) use the meta-regression model built to predict ecosystem service value [27]. Meanwhile, there are very few applied research on meta-analysis value transfer method. According to the current literature retrieval condition, overseas and domestic scholars have already launched the applied research of meta-analysis value transfer method in China’s forest ecosystem service value assessment [28], study of dry farmland ecosystem’s active and passive restoration measures [29], land use changes and ecosystem service value transfer in Changbai Mountains region [30], study of agroforestry system’s grain yield [31], and assessment of wetland ecological protection value in Changbai Mountain region [32]. However, there is no applied research of farmland ecosystem service value assessment in China yet.

IPCC SRES has four broad scenarios: A1, A2, B1, and B2. Scenario A describes a world that tends toward economic development, while Scenario B tends toward social and environmental development; Scenario 1 has a more globalized world with frequent economic, cultural, and technological interactions among countries, while Scenario 2 has a less globalized world. Specifically, the A1 scenario depicts a future that is called the “Golden Age of the Economy. Scenario A2 is called “Cultural Diversity”. People are concerned with regional independence and environmental issues are not a priority. Populations continue to grow, trade barriers exist, and technological progress is slow. Scenario B1 is similar to “sustainable development”: there is a high level of globalization, stable populations, international cooperation, a preference for more efficient and cleaner energy sources, and rapid development of energy-efficient technologies. “Regional Solutions”. People are concerned about regional independence, but also about environmental protection. Some regions would use clean coal as fuel, others would use clean energy sources such as wind and solar. Among them, A1B fuel use is more balanced. The different scenarios are based on different drivers, such as economy, energy use and temperature. Several specific drivers are represented in these four scenarios:Economic development: globalization is better than non-globalization (1 series > 2 series), but focusing on the economy does not necessarily yield better benefits than focusing on the society/environment (A2 < B2).The Human Development Index (HDI) is an index proposed by the United Nations Development Programme (UNDP) that includes three dimensions: human health and longevity, knowledge and education, and economic development. A2 is basically the worst scenario, with B2 slightly worse.Temperature rise: Focus on more economic warming (A > B), less globalization warming (1 < 2)

In light of the above problems, we collected the value assessment results of empirical research literature about farmland ecosystem service value assessment in China, established a value transfer database, built a meta-analysis value transfer model of farmland ecosystem service in China by using meta-analysis and multiple regression analysis method, and calculated farmland ecosystem value change condition in China between 2010 and 2100 based on these. First, an influential factor analysis and a model error evaluation were conducted though the regression results. Then, on this basis, according to the A1B, A2, B1, and B2 scenarios of IPCC SRES, the farmland value change condition in China between 2010 and 2100 was calculated, and the applicability and development prospect of Meta-analysis value transfer method in farmland ecosystem service value assessment field in China were discussed.

## 2. Research Method and Data Processing

### 2.1. Database Establishment

The monetizing valuation of the entire biosphere’s ecosystem service value made by Costanza et al. [33] has become an important milestone for the entire ecosystem service value assessment system in future. Now, there are three sources for ecosystem service value in China: Costanza et al. [33], The Millennium Ecosystem Assessment (MA) [34], and Xie Gaodi et al. [35]. This paper referred to the value assessment procedure of existing literature and the MA reports, and established a meta-analysis farmland ecosystem service value assessment system (see Figure 1).

Based on the farmland ecosystem service value assessment system in Figure 1, the research information such as the title, author, research time, research region, farmland type, evaluation method, ecosystem service type, and ecosystem service value were collected uniformly from the literature and then input into the Excel table. At last, a total of 26 literature and 70 value observations were included in the meta-analysis database. It can be seen from the geographical location distribution of study sample site (Figure 2) that the evaluated farmland ecosystems were mainly distributed in the major grain-producing area such as Huanghai-Huaihai-Haihe region and Yangtze Plain, Middle, and Lower, while very few study samples were distributed in the non-major grain-producing area such as the central and western regions of China.

### 2.2. Selection of Independent Variables

The study on ecosystem service value transfer at home and abroad generally considers the value assessment method of study samples, research region area, ecosystem service type, regional economic development level of research region, substitution effect of the same type of ecosystems, as well as the influence of other factors on ecosystem service value changes [36]. By reference to overseas and domestic influential factors considered, combined with the actual data collection condition in this paper, the independent variables used to build meta-regression model were selected:

Ecosystem service type: different types of services provided by an ecosystem are different, their importance are also differed, so the generated values are varied. The difference between different ecosystem service types may have impact on the value of farmland ecosystem. 

Value assessment method: the ecosystem value assessment method can affect the assessment results to a great extent, especially the selection of service quality evaluation index and price parameters. This paper focuses on analyzing the influence of using different value assessment methods on the change of farmland ecosystem service value. 

Farmland division: farmland is mainly divided into major crops producing region and non-major crops producing region. The crops producing region covers the northeast plain, Huanghai-Huaihai-Haihe plain, Yangtze River plain, with wide distribution scope and good production condition.

Farmland area: it is known from recently collected literature that ecosystems have boundary effect. Within the boundary effect, with the increase of farmland area, the unit area value of some farmland ecosystems will increase as well; but when it exceeds a certain critical threshold, the value of some farmland ecosystems may show decreasing scale benefits.

Number of beneficiaries: the number of beneficiaries of farmland ecosystem service reflects the demands or market size of farmland ecosystem service [37]. This paper defines the scope of farmland beneficiaries as within the administrative region. The population data within administrative regions originates from the Statistical Yearbook of each province and city.

Per capital GDP: the farmland ecosystem service value is closely related to the economic development level of the region in which the farmland is located, for example, there is a huge gap between the developed and under-developed regions in identification and market realization of farmland ecosystem service value [38]. This paper uses per capital GDP as the index to measure regional economic development state. According to data collection condition, the scale of the region is defined as within the administrative region. The per capital GDP data of the administrative region in which the farmland is located originate from the Statistical Yearbook of each province and city.

Cultivated land type: the level-1 cultivated land mainly includes paddy field and dry land. Dry land normally refers to the lands in which dry crops are planted without seasonal irrigation. Paddy field refers to the farmlands in which paddy and other aquatic plants are planted with seasonal ponding every year. Different cultivated land types may have impact on farmland ecosystem service value.

Count the number and mean of value observations corresponding to each variable, follow the statistical and metric data requirements to assign various information of independent variables, and calculate their mean values and standard deviations respectively. See the variable information of meta-regression model in Table 1 [3,14,25,39,40,41,42,43,44,45,46,47,48,49,50,51,52,53,54,55,56,57,58,59,60]. 

### 2.3. Model Establishment

In meta-regression, the weighted regression model, panel data model or hierarchical linear model are usually used to explain individual research effect [61]. In weighted regression, every study has its independent weight, irrelevant to the number of value obtained from a single research. Panel data include fixed effect model and random effect model, in which the former assumes every study has fixed individual effect, and its interior estimator could only explain the influence inside the study. While the latter assumes every study has random individual effect, and it often uses the generalized least square estimation, able to explain the influence between studies and inside the study simultaneously. Hierarchical linear model mixes random and fixed models [62]. However, in meta-regression dataset of this paper, the estimated value variance change is not sufficiently explained inside single research, so panel data model or hierarchical model is not suitable for the dataset of this paper [62]. 

In the existing relevant meta-regression analysis, the majority adopts the ordinary least squares(OLS), with general expression as below: (1)$$\ln\;(y_{i})=\alpha +\beta_{s}X_{s}+\beta_{t}X_{t}+\beta_{p}X_{p}+\beta_{e}X_{e}+\delta_{i}$$

In which, the dependent variable y is the value vector of farmland ecosystem service, in CNY ha^−1^ a^−1^. α is a constant term, *δ* is a residue term, *β* is the regression coefficient matrix of independent variable, and *X* is the independent variable matrix, in which, *X_s_* represents the variable of farmland ecosystem service value assessment method, *X_t_* represents the characteristics of evaluated farmland, *X_p_* represents farmland ecosystem service type, *X_e_* represents the geographical environment characteristics around evaluated farmland.

The main reason for choosing a log-linear model in this paper is that the logarithmic transformation can reduce fluctuations in the original data, improve the accuracy of the fit and reduce heteroskedasticity [62,63]. Since the value assessment in literature was often based on different years, in order to make the data comparable, the paper used the value observation conversion method adopted by Kochi et al. [64] and Johnston et al. [65] for reference, and regulated the values of different assessment base years into the price level of 2015 through consumer price index (CPI); then, divided the value (CNY a^−1^) of unified base year by farmland area (ha) of study region, obtained the unit area value (CNY ha^−1^ a^−1^) of farmland ecosystems in different study regions, and took it as the dependent variable of meta-regression model. 

## 3. Results Analysis

### 3.1. Meta-Regression Analysis

The meta-regression results are shown in Table 2. The Model(A) reports the general model covering all explanatory variables and makes estimation through OLS, but the B-P test results indicate the existence of heteroscedasticity (*p* = 0.000). The reason may be that the potential assumption of least square method is that different observations are irrelevant. In the established meta-analysis database, one literature can provide 11 value observations at the most, and 50% literature provide multiple observations. Since the observations from the same literature are not independent and different studies may be correlated, these could result in biased estimation simultaneously. In Model(B), we referred to the solutions to this problem in existing studies and used the weighted least square (WLS), in which some used the weighted least square taking the reciprocal of observation number as the weight [36,63,66], reducing the influence of sample correlation to a certain extent. But the demerit of WLS is that it assumes the covariance matrix of disturbing term is known, which is often an unrealistic hypothesis. In view of this, it only can be used after using sample data for uniform estimation, and this method is called the feasible weighted least square (FWLS) [67], see the results in Model (C). Moreover, we tested whether the regression model had serious multicollinearity problem by figuring out the variance inflation factor (*VIF*), which was defined as:(2)$$VIF=\frac{1}{1-R_{i}^{2}}$$

In which, *R*_i_ is the negative correlation coefficient of regression analysis performed by independent variable X_i_ on other independent variables. The greater the *VIF* is, the larger the possibility of multicollinearity exists between independent variables will be. Generally speaking, if *VIF* is more than ten, the regression model has serious multicollinearity. While the independent variable’s VIF being less than ten is generally acceptable, this indicates there is no multicollinearity problem between independent variables [67]. It is known from Table 3 that the VIF of the independent variable-number of beneficiaries is greater than ten, so after we delete it from Model(D-G), the VIF values of all independent variables are tested to be less than ten. It can be seen from the results of Model (D) that the significance of farmland area has been improved, and the independent variable-number of beneficiaries is not significant in Model (A) and Model (B). This demonstrates that the model accuracy has been improved after the number of beneficiaries is deleted. In Model (E), the significance of opportunity cost method and per capital GDP declines slightly, but it is still statistically significant at a level of 5%, but the significance of such ecosystem service type variables as crops and gas regulation rise to some extent. This may be due to the influence of multicollinearity interference which can be improved after deleting the influential variables. Brander et al. [68] considers when evaluating a given ecosystem’s service value, more than one assessment method is often used, and the value assessment method of a given ecosystem service is generally definite, for example, using shadow project approach to evaluate the value of water conservation [69], using opportunity cost method to evaluate the value of biodiversity, etc. [70] Therefore, the regression results can be affected. So, we deleted the value assessment method variables from Model (F) and Model (G), but its results showed that except the significance of farmland area did not change, the significance and regression coefficient of other variables had changed. This phenomenon also manifests that the value assessment method play a certain effect in regression model, so we should refer to existing studies to reserve the value assessment method variable. [36,64,66].

In Model (E), the ecosystem service type, value assessment approach, farmland division, cultivated land type, farmland area and per capital GDP in all can explain 98.3% of the value change of sample size farmland. In the regression results, the regression coefficient of virtual variables (ecosystem service type, value assessment method, farmland division, cultivated land type) reflects the deviation direction and degree of specific variables relative to control group; the regression coefficient of continuous variable represents elastic coefficient, i.e., the ratio of change rate of dependent variable to independent variable. A specific analysis of regression results is made as below: Value assessment method: the regression coefficient of opportunity cost approach and market evaluation approach is statistically significant. This indicates when other influential factors remain unchanged, the value estimates obtained through opportunity cost approach and market valuation approach show significant difference with that obtained through surrogate market approach. The market valuation approach is higher than other value assessment method, while the value estimated obtained through opportunity cost approach is the lowest.Ecosystem service type: in the six farmland ecosystem services, the regression coefficients of crops, gas regulation, soil conservation, and recreational tourism are statistically significant. This indicates when other influential factors remain unchanged, the values of crops, gas regulation, soil conservation, and recreational tourism services show significant difference with water conservation. The values of crops, gas regulation, and soil conservation are higher than other farmland ecosystem services, in which the value of soil conservation service is the highest, while the value of recreational tourism service is the lowest.Cultivated land type: the regression coefficient of paddy field is positive and is statistically significant at the level of 1%. This indicates when other influential factors remain unchanged, the value estimates obtained when farmland ecosystem’s cultivated land type is paddy field are evidently higher than that obtained from dry land.Farmland division: in the four farmland division regions, the regression coefficient of Yangtze Plain Middle and Lower is statistically significant. This indicates when other influential factors remain unchanged, the farmland ecosystem service value of Yangtze Plain Middle and Lower is obviously lower than the service value of other regions.Farmland area: the regression coefficient of farmland area is significantly negative. This indicates the per hectare value of farmland ecosystem has decreasing return to scale, but this effect will decrease geometrically with the increase of ecosystem area [71,72]. Taking the regression coefficient −0.181 of farmland area as an example, for a farmland ecosystem of 10 ha, if the area increases by 1%, its per hectare value will decrease by 1.81%; but for a farmland ecosystem of 1000 ha, if the area increases by 1%, its per hectare value only decreases by 0.018%. So, the total value of farmland ecosystem still increases with the increase of farmland area.Per capital GDP: the regression coefficient of per capital GDP variable is statistically significant, showing negative correlation. This indicates when other conditions remain unchanged, if the per capital GDP of research region becomes higher, the unit area value of farmland ecosystem will be lower, and economic growth may lead to a recession of ecosystem function. When per capital GDP increases by 10%, the unit area value will reduce by 0.9%.

### 3.2. Value Transfer

Transfer error is used to test the consistency between model prediction value and value observations [12,28], equivalent to the mean absolute percentage error (*MAPE*), defined as:(3)$$MAPE=\sum [\frac{{|V}_{\mathrm{est}}-V_{\mathrm{obs}}|}{V_{\mathrm{obs}}}\cdot 100]/n$$ where V_est_ is the transferred (predicted) farmland value from the MRA, V_obs_ is the farmland value as reported in a primary study, and *n* is the number of estimates. Generally speaking, the smaller the transfer error is, the higher the effectiveness of value transfer model will be [12,28].

The paper adopts the leave-one-out cross validation [63,68,73], i.e., successively select every observation as test set, take other observations as training set, and calculate the transfer error of corresponding observation in the test set respectively. Compared to the ordinary K-fold cross validation(k > 1), though the calculation of leave-one-out cross validation is more tedious, its sample use ratio is the highest. It not only can accurately evaluate the model’s global error, but also make it easier to analyze the change characteristics of error with observations.

We use MAPE to calculate the transfer error rate according to Equation (3). The transfer error rates are presented in Table 3, columns (1) and (2) for in-sample comparisons. Column (1) represents the transfer error of Model (D) in Table 2, and column (2) represents the transfer error of Model (E), with the average transfer errors of 47.60% and 36.74%, respectively. The results show that the regression results of FWLS are better than WLS results. We also report out-of-sample predictions in columns (3) and (4), with average errors of 94.96% and 88.87%, respectively, both of which are greater than in-sample errors. This condition conforms to the prediction [71]. The column (2) in Table 3 indicates that the median transfer error rate is 14.59%, in which, 74% of sample transfer error rate is less than 40%, and 9% of sample transfer error rate is greater than 100%. Therefore, it is necessary to be cautious due to that some transfer error rates are very large.

By sorting the value observations in ascending order, the change condition of model’s predicted value and transfer error is obtained as shown in Figure 3 and Figure 4. Figure 3 displays the change of predicted values of Model (D) and Model (E). It can be seen that when value observations are small, the model’s predicted value is higher with large deviation degree; when value observations are large, the model’s predicted value is lower with gradually descending deviation degree and data fluctuation range. The predicted value deviation and fluctuation range of Model (E) is obviously better than that of Model (D). Figure 4 indicates that with the increase of value observations, the transfer error shows a declining trend, and the transfer error of Model (E) is less than the results of Model (D) in most cases.

### 3.3. Value Change Assessment of China’s Farmland Ecosystem between 2010 and 2100

This paper adopts the global 1 km land cover change dataset of 2010~2100 established by Li Xia et al. [2]. This dataset is built based on IMAGE module [74] and cellular automata techniques [75], and simulates the evolutionary process of different land use types between 2010 and 2100 according to the four scenarios of IPCC SRES: A1B, A2, B1, and B2, in which the baseline scenario is in the year of 2010. A1B belongs to the sub-scenario of various energy balance development in A1 scenarios. This dataset can be downloaded at https://geosimulation.cn/download/GlobalSimulation/ (accessed on 2 December 2021).

The paper extracted the dataset of China’s land use (Farmland, Water, Urban, Forestry, Barren and Grassland) in the years of 2050 and 2100 under four scenarios, as shown in Figure 5. The meta-regression model results were used to calculate the change of China’s farmland ecosystem service value under different scenarios, see the calculation results in Table 4. In 2010, the total farmland area in China was about 110 million hectare, and the total ecosystem service value was 8860 billion yuan. Under four scenarios, the change in total values of farmland area and farmland ecosystem service with time shows the same tendency. Under A2 and B2 scenarios, the total values of farmland area and farmland ecosystem service continuously increase, reaching to 15,220 billion yuan and 14,930 billion yuan till the year of 2100, respectively. Under A1B and B1 scenarios, both the total values of farmland area and farmland ecosystem decline after a rise, and the total farmland values will be only 7920 billion yuan and 6320 billion yuan in the year of 2100, respectively. In general, as China’s farmland ecosystem is concerned, A2 scenario is the optimal development path, and the total service value loss of farmland ecosystem in B1 scenario is the maximum.

## 4. Discussion

It is worth noting that meta-analysis is a method of performing quantitative comprehensive analysis and variation source analysis on existing results, so the effectiveness of meta-analysis-based value transfer method depends on the quantity and quality of existing empirical study to a great extent. But at present, there are a few research on farmland ecosystem in China, which may affect the research results. So, we suggest to further improve the quantity and quality of relevant empirical studies and attach importance to the technical perfection and index integrity in the process of ecosystem service value assessment. Meanwhile, it is also found from this study that whether urban development or national development in future, it is essential to pay attention to protecting arable land minimum and persist in permanent basic farmland policy, which could bring huge benefit for numerous natural ecosystems and social ecosystems.

## 5. Conclusions

China has always been a great agricultural country, in which farmland ecosystem development directly relates to national development. In this paper, MRA was made on the service value of farmland ecosystem in China, providing a novel contribution to the study of farmland ecosystem service value transfer. We discover that farmland characteristics, assessment method, and service type all affect the value of farmland. For example, farmland area negatively affects farmland value, and the farmland with cultivated land type as paddy field is more valuable than that as dry land. In addition, the farmland that provides crops and that has good soil conservation function is more valuable than that used for recreational tourism. This demonstrates that converting farmland into tourism development may bring down its value, but protecting it may increase its value. Another result is that the farmland value evaluated by using market valuation method is obviously superior to other value assessment methods, so the market valuation approach should be preferred in the farmland ecosystem service value assessment method in future. By sorting the value observations in ascending order, it is noted that with the increase of value observations, the model predicted value gradually transits from overestimation to underestimation, and transfer error also tends to decrease. This result is consistent with the research results of Brander et al. wetland ecosystem) and Salem et al. (mangrove forest ecosystem). Recently, there is still no reasonable explanation to such phenomenon, but it can be taken as a reference for value transfer process.

Regarding value transfer model construction, compared to the published meta-analysis studies, this paper used fewer explanatory variables, without considering such influential factors as the physical characteristics, environmental quality, climate change, and human activities of farmland ecosystem. The reason is the lack of locating observation and experimental research of ecosystem function in the study of farmland ecosystem in China, as well as the formation process of various ecosystem services and the mechanism study of human influence, so it is difficult to include relevant influential factors in this meta-regression analysis model. This is also a regret of this study. Regarding the results, we used the leave-one-out cross validation to obtain the error range of 88.87% and its median of 23.04%, both of which are higher than in-sample error by 36.74% and 16.59%. The FWLS error fluctuation results are significantly better than the WLS model. The studies on value transfer of farmland ecosystem in China between 2010 and 2100 indicate that the change trend in total values of farmland area and ecosystem is consistent, conforming to the opinions of Brander et al. and Woodward et al. However, the specific change conditions are somewhat varied under different scenarios, in which the total value increases evidently under A2 scenario, while the total value loss is the maximum under B1 scenario.

## Figures and Tables

**Figure 1 ijerph-20-00440-f001:**
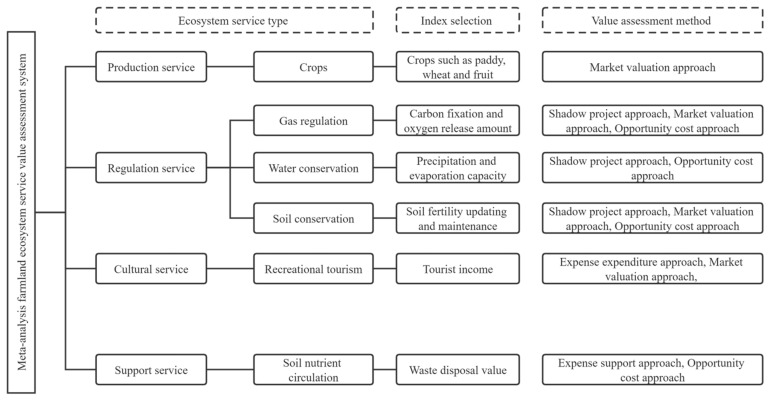
Meta-analysis farmland ecosystem service value assessment system.

**Figure 2 ijerph-20-00440-f002:**
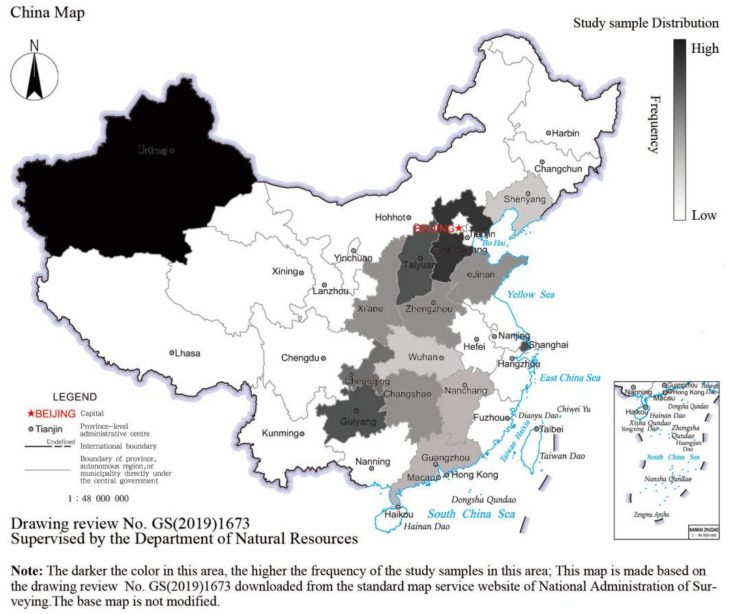
Geographical location distribution of study samples.

**Figure 3 ijerph-20-00440-f003:**
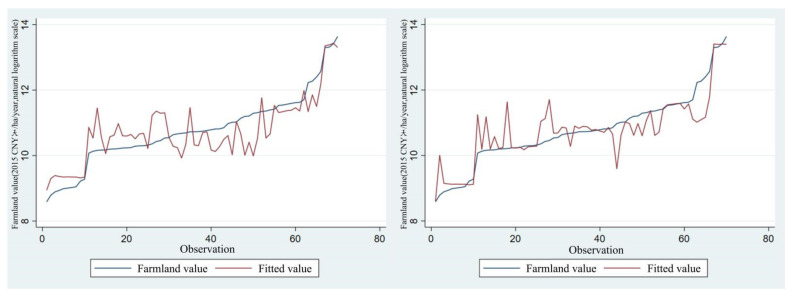
Value observations and predicted value (value observations are sorted in ascending order, with Model (D) on the left and Model (E) on the right).

**Figure 4 ijerph-20-00440-f004:**
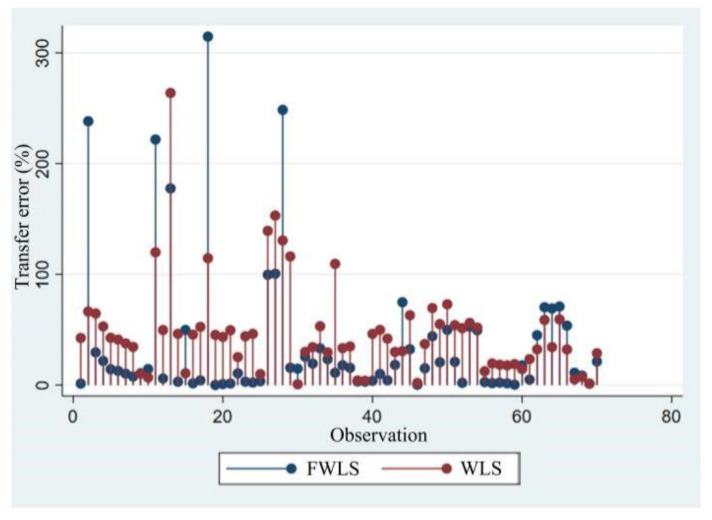
FWLS and WLS transfer error.

**Figure 5 ijerph-20-00440-f005:**
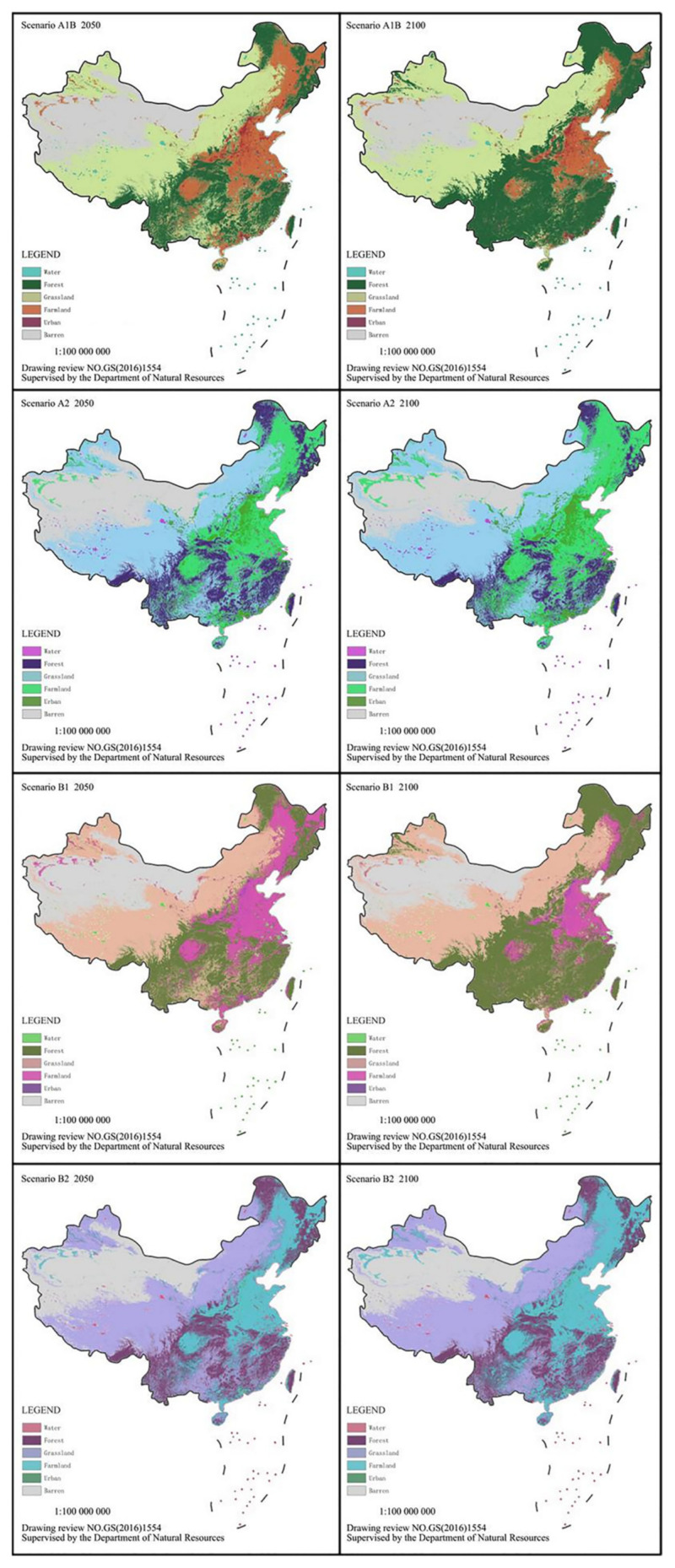
China’s land use distribution in different scenarios (with no modification to base map).

**Table 1 ijerph-20-00440-t001:** Variable definitions and summary statistics ^a^ (in CNY¥ ha^−1^ yr^−1^).

Variable Names	Variable Description	Mean	SD	N
Dependent variable				
Farmland value	Annual value per hectare in 2015 CNY¥ in logarithmic form	10.770	1.101	70
Independent variables				
Value evaluation approach				
Surrogate market approach	Baseline category ^b^	0.143	0.352	10
Opportunity cost approach	If the opportunity cost approach is used for assessment, the value is set as 1, otherwise as 0	0.443	0.500	31
Carbon tax approach	If the carbon tax approach is used for assessment, the value is set as 1, otherwise as 0	0.143	0.352	10
Shadow project approach	If the shadow project approach is used for assessment, the value is set as 1, otherwise as 0	0.457	0.502	32
Market valuation approach	If the market valuation approach is used for assessment, the value is set as 1, otherwise as 0	0.529	0.503	37
Farmland ecosystem services				
Water conservation	Baseline category ^b^	0.500	0.504	35
Crops	If the ecosystem service type is crops, the value is set as 1, otherwise as 0	0.786	0.413	55
Gas regulation	If the ecosystem service type is gas regulation, the value is set as 1, otherwise as 0	0.600	0.493	42
Soil conservation	If the ecosystem service type is soil conservation, the value is set as 1, otherwise as 0	0.457	0.502	32
Recreational tourism	If the ecosystem service type is recreational tourism, the value is set as 1, otherwise as 0	0.257	0.440	18
Soil nutrient circulation	If the ecosystem service type is soil nutrient circulation, the value is set as 1, otherwise as 0	0.414	0.496	29
Dry land	Baseline category ^b^	0.643	0.483	45
Paddy field	If the cultivated land type is paddy field, the value is set as 1, otherwise as 0	0.371	0.487	26
Non-major crops producing region	Baseline category ^b^	0.700	0.462	49
Northeast region	If the crops’ geographic region is northeast region, the value is set as 1, otherwise as 0	0.043	0.204	3
Huanghai-Huaihai-Haihe region	If the crops’ geographic region is Huanghai-Huaihai-Haihe region, the value is set as 1, otherwise as 0	0.257	0.440	18
Yangtze Plain, Middle and Lower	If the crops’ geographic region is Yangtze Plain, Middle and Lower, the value is set as 1, otherwise as 0	0.057	0.234	4
Farmland size	Area of Farmland site in logarithmic form	12.599	2.256	70
Number of beneficiaries	Numerical variables in logarithmic form	15.476	2.485	70
GDP per capita ^c^	GDP per capita in logarithmic form	9.896	0.961	70

^a^ Note: N = number of observations for each variable or variable level; SD = standard deviation. ^b^ Baseline category refers to that which is excluded for each categorical variable in order to avoid perfect collinearity. ^c^ Referring to year 2015.

**Table 2 ijerph-20-00440-t002:** Estimated meta-regression value transfer function.

Variable	Full Model			Reduced Model			
	Model (A)	Model (B)	Model (C)	Model (D)	Model (E)	Model (F)	Model (G)
Opportunity cost approach	−0.462 (0.296)	−0.288 (0.239)	−0.559 *** (0.194)	−0.326 (0.230)	−0.465 ** (0.190)	-	-
Carbon tax approach	−0.279 (0.318)	−0.170 (0.374)	0.068 (0.422)	−0.208 (0.395)	0.116 (0.358)	-	-
Shadow project approach	0.344 (0.294)	0.189 (0.257)	−0.084 (0.209)	0.262 (0.256)	−0.171 (0.184)	-	-
Market evaluation approach	0.870 *** (0.215)	1.326 *** (0.262)	0.623 *** (0.134)	1.338 *** (0.257)	0.627 *** (0.128)	-	-
Crops	0.224 (0.256)	0.443 * (0.257)	0.445 ** (0.168)	0.403 * (0.227)	0.464 *** (0.168)	0.164 (0.348)	0.403 * (0.225)
Gas regulation	0.237 (0.297)	0.212 (0.331)	0.490 (0.323)	0.188 (0.336)	0.495 * (0.284)	0.417 (0.367)	0.486 ** (0.189)
Soil conservation	0.444* (0.230)	0.152 (0.222)	0.872 *** (0.164)	0.169 (0.216)	0.923 *** (0.155)	0.411 (0.397)	1.577 *** (0.297)
Recreational tourism	−0.780 *** (0.249)	−0.654 ** (0.273)	−0.989 *** (0.343)	−0.629 ** (0.246)	−1.018 *** (0.301)	−0.540 (0.391)	−0.544 ** (0.240)
Soil nutrient circulation	0.255 (0.253)	0.403 (0.270)	−0.108 (0.230)	0.409 (0.269)	−0.152 (0.209)	−0.273 (0.294)	−0.419 *** (0.142)
Paddy field	0.734 *** (0.240)	0.996 *** (0.330)	0.649 *** (0.222)	1.013 *** (0.323)	0.590 *** (0.199)	0.747 * (0.444)	0.080 (0.315)
Northeast region	−0.222 (0.453)	−0.268 (0.685)	0.103 (0.403)	−0.220 (0.665)	−0.073 (0.403)	−0.368 (0.855)	−0.841 * (0.493)
Yangtze Plain, Middle and Lower	−1.056 *** (0.331)	−1.411 *** (0.360)	−1.147 *** (0.412)	−1.411 *** (0.357)	−1.141 *** (0.371)	−0.921* (0.544)	0.321 (0.528)
Huanghai-Huaihai-Haihe region	−0.388 (0.269)	−0.422 (0.418)	−0.189 (0.359)	−0.407 (0.422)	−0.218 (0.303)	−0.407 (0.396)	0.080 (0.248)
Farmland area	−0.215 ** (0.082)	−0.166 (0.104)	−0.149 *** (0.053)	−0.135 ** (0.053)	−0.181 *** (0.046)	−0.190 *** (0.067)	−0.298 *** (0.044)
Number of beneficiaries	−0.014 (0.087)	0.035 (0.099)	−0.034 *** (0.006)	-	-	-	-
Per capital GDP	−0.083 (0.090)	−0.149 * (0.075)	−0.106 *** (0.034)	−0.149 * (0.074)	−0.090 ** (0.035)	−0.040 (0.125)	0.047 (0.052)
Constant term	13.620 *** (1.112)	12.420 *** (0.853)	13.268 *** (0.357)	12.570 *** (0.771)	13.014 *** (0.323)	13.129 *** (1.669)	13.108 *** (0.640)
Number of observations	70	70	70	70	70	70	70
R^2^	0.827	0.742	0.990	0.741	0.987	0.554	0.937
Adjusted R^2^	0.775	0.664	0.986	0.670	0.983	0.469	0.925

Note: Figures in brackets are standard errors. Model (A) uses OLS. WLS is used for Model (B, D, F), using reciprocal of sample size as weights. FWLS is used for Model (C, E, G). Significance is indicated with ***, **, and * for 1%, 5%, and 10% statistical significance levels, respectively.

**Table 3 ijerph-20-00440-t003:** MAPE of MRA models.

MAPF (%)	In-Sample MAPE		Out-of-Sample MAPE	
	(1) Model (D)	(2) Model (E)	(3) Model (D)	(4) Model (E)
Average MAPE	47.60	36.74	94.96	88.87
Median MAPE	42.18	14.59	47.97	23.04
Maximum	263.69	314.64	874.91	1802.72
Minimum	0.72	0.07	0.84	0.04

Note: Columns (1) and (3) use the WLS results from Model (D), Table 2. Columns (2) and (4) use the FWLS results from Model (E), Table 2. Columns (1) and (2) report the in-sample MAPE. Columns (3) and (4) report out-of-sample comparisons.

**Table 4 ijerph-20-00440-t004:** Change in farmland area and value in China under different scenarios (2010–2100).

Scenarios	Year	Area (10^8^ ha)	Value (10^12^ CNY)
Baseline Scenario	2010	1.10	8.86
A1B Scenario	2050	2.29	14.86
	2100	0.96	7.92
A2 Scenario	2050	1.93	13.44
	2100	2.40	15.22
B1 Scenario	2050	1.50	11.31
	2100	0.74	6.32
B2 Scenario	2050	1.76	12.66
	2100	2.31	14.93

Note: A1B Scenario: low-speed population growth; sprawling city; super high-speed economic growth; rapid technological innovation; strong bio-fuel demand, balanced development of various energy. A2 Scenario: high-speed population growth; sprawling city; medium-speed economic growth; slow technological innovation; lower bio-fuel demand. B1 Scenario: low-speed population growth; compact city; high-speed economic growth; slower technological innovation; low overall energy consumption, low bio-fuel demand. B2 Scenario: medium-speed population growth; compact city; medium-speed economic growth; slower technological innovation; less overall energy consumption, low bio-fuel consumption.

## Data Availability

Not applicable.

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
