# Peer review of "Study on Chinese Farmland Ecosystem Service Value Transfer Based on Meta Analysis"

_ijerph, 2022, doi:10.3390/ijerph20010440_

Round 1
Reviewer 1 Report
I got a pleasure during the reading process of the article. However, I feel that the research needs major revision.
1. It is not clear which three criteria are referred to in lines 81-82. It seems that we are talking about sources, not criteria... After the MA, there were other projects and studies on the valuation of ecosystem services, in particular TEEB and others. The authors almost completely ignore other studies, taking into account only works from China.
2. It is not clear why the authors limited themselves to only 6 ecosystem services (Figure 1). It can be seen that the origins of the classification are taken from MA, but it is not clear how the authors substantiate this. A more modern classification is CICES, there are also not to mention other private and TEEB’ classifications.
3. I opened the link https://geosimulation.cn/download/GlobalSimulation/, but when downloading and opening the file for each scenario, I see a black screen.
Reviewer 2 Report
Please consider the following suggestions:
Lines 71-75: In my opinion, this paragraph needs some background information on IPCC SRES. Please briefly explain the meaning of these abbreviations and include some supplemental information about the Panel and its purpose in assessing the risk of human-induced environmental change. This information could greatly improve the quality of your manuscript, as the core of the results presented relate to the implementation of SRES scenarios in predicting different futures of farmland ecosystem development.
Lines 222-224: Citations must meet journal standards. The Brander et al. reference (line 222) should be cited as Brander et al. [72].
Line 364: Conclusions and Discussion: Discussion should be separated from the Conclusions.
Reviewer 3 Report
The manuscript entitled "Study on Chinese Farmland Ecosystem Service Value Transfer Based on Meta Analysis " intends to identify the influence factors that triggered the changes of Chinese farmland ecosystem values following a literatures review. However, this manuscript has very poor quality, in which the authors had used wrong or at least incomplete methods. Despite literatures review, the authors did not know well about the theme---farmland ecosystem services, and the manuscript even lacks deeper analysis and illustration on the literatures acquired by themselves. In addition, the language should be extensively edited while the structure should be re-organized. Overall, I strongly suggest to reject this manuscript. At present, the priorities for it should re-organize the structure, provide some supplements on literatures review as well as polish the text throughout the manuscript, rather than focusing on some details.
Round 2
Reviewer 1 Report
Almost all comments were taken into account.
Author Response
Thank you for your comments.
Reviewer 3 Report
The authors for this revision version have made many efforts. At present, I think this manuscript can be published following some revisions in text editing.
